# Automatic Obstacle Detection Method for the Train Based on Deep Learning

**Qiang Zhang** [1] , **Fei Yan** [1,*] , **Weina Song** [1], **Rui Wang** [2] **and Gen Li** [1]

1 School of Electronic and Information Engineering, Beijing Jiaotong University, Beijing 100044, China
2 School of Computer and Information Technology, Beijing Jiaotong University, Beijing 100044, China
* Correspondence: fyan@bjtu.edu.cn

**Abstract:** Automatic obstacle detection is of great significance for improving the safety of train operation. However, the existing autonomous operation of trains mainly depends on the signaling control system and lacks the extra equipment to perceive the environment. To further enhance the efficiency and safety of the widely deployed fully automatic operation (FAO) systems of the train, this study proposes an intelligent obstacle detection system based on deep learning. It collects perceptual information from industrial cameras and light detection and ranging (LiDAR), and mainly implements the functionality including rail region detection, obstacle detection, and visual–LiDAR fusion. Specifically, the first two parts adopt deep convolutional neural network (CNN) algorithms for semantic segmentation and object detection to pixel-wisely identify the rail track area ahead and detect the potential obstacles on the rail track, respectively. The visual–LiDAR fusion part integrates the visual data with the LiDAR data to achieve environmental perception for all weather conditions. It can also determine the geometric relationship between the rail track and obstacles to decide whether to trigger a warning alarm. Experimental results show that the system proposed in this study has strong performance and robustness. The system perception rate (precision) is 99.994% and the recall rate reaches 100%. The system, applied to the metro Hong Kong Tsuen Wan line, effectively improves the safety of urban rail train operation.

**Keywords:** rail area detection; obstacle detection; rail traffic; deep convolutional neural network

## 1. Introduction

With the rapid development of rail transit, fully automated operations systems step into a stage of rapid development [1,2]. The standard "Railway applications Automated urban guided transport Safety requirements Part I: General" (GB/T 32588.1-2016) [3] issued in 2016 clearly states that in driverless train operation (DTO) and unattended train operation (UTO) operation modes, there will be no operating staff in the front-end cabin of the train. The control system is responsible for performing rail area detection and preventing the risk of collision with obstacles within the line boundaries [4,5]. Therefore, the operational safety of fully automatic operation (FAO) systems requires the support of the automatic environmental perception of trains.

The obstacles which may appear on the rail lines, such as pedestrians, falling rocks, and intruders, would break the safety conditions of rail line closure. Thus, detecting these obstacles and reducing the probability of collision accidents are the goals of the intelligent train detection system [6].

With the maturity of urban rail transit technology, intelligent train detection systems have become an important development direction. The intelligent train detection system covers modern data communication, automatic control, highly reliable positioning, and other advanced technologies, and represents the highest level of rail transit modernization. First, the intelligent train detection system is not affected by the subjective factors of drivers; it is of great significance to improve the safety of vehicle operation [7–9]. Second, compared with the

traditional manual driving train, the intelligent train detection system will surely improve the operation efficiency. It has strong flexibility and adaptability in the supply and configuration of transportation services, which effectively ensures the punctuality and comfort of train operation. It also helps to enhance the service quality of the transportation system.

At present, there are many schemes of intelligent detection systems applied in the traffic field. These mainly include the detection system scheme based on computer vision, radar (millimeter-wave radar), and visual–LiDAR (Light Detection and Ranging) fusion. The detection scheme based on computer vision collects the images from the moving direction of the train through cameras [10]. This method only requires the configuration of the camera sensor and does not require complicated calibration [11,12]. This solution is economical, simple to deploy, and can be employed in a dark environment. However, the camera is greatly affected by the external environment (such as light, rain, and snow). The detection scheme based on radar mainly transmits and recovers the signal through the radar and compares the received signal frequency with its frequency to obtain the relative distance and speed between the vehicle and the obstacle [13]. Radar has the characteristics of small volume, light weight, and high spatial resolution. Meanwhile, the millimeter wave guide seeker has a strong ability to penetrate fog, smoke, and dust. However, this solution requires the deployment of equipment beside the railway, which increases the difficulty of implementation and maintenance. The detection scheme based on visual–LiDAR fusion integrates the information of visual and LiDAR [14]. It can make up for the deficiencies of a single vision or LiDAR detection schemes. The scheme is less affected by environmental factors, so it has high reliability.

In this paper, the key methods of the intelligent train detection system are studied. We integrate visual and LiDAR information, which realizes pixel-wise identification of the rail track area and obstacles in front of the running train. The FAO system can leverage the perceptual information to warn the control center or directly output a brake. It ensures that the train can accurately identify the obstacles ahead and enhances the safety of train operation.

The general contributions of this work are as follows:

(1) The convolutional neural networks are leveraged to realize the computer vision tasks under the urban railway scenario, i.e., rail track area detection and obstacle detection;

(2) A visual and LiDAR information fusion method is proposed. It achieves a better performance for obstacle detection by implementing the two aforementioned tasks in real-time;

(3) A method to judge the relationship between rail track and obstacles is proposed to improve the accuracy of early warning.

## 2. Literature Review

### 2.1. Rail Track Region Detection

The existing rail track region detection algorithms are mainly divided into two categories. One is to extract the rail region by feature extraction operator, the other is to detect the rail region based on deep learning method.

Qi et al. [15] propose to use gradient direction histogram features to build integral images and use the area growth algorithm to detect the rail area. Based on the prior knowledge, Wu et al. [16] preliminarily determine the characteristics of the rail in the lower part of the image and then detect the rail area through the angle calibration measurement algorithm. This algorithm shows strong robustness to both color and light. Researchers [17,18] used a multi-threshold algorithm to segment rail tracks in a complex environment. They used the pixel tracking algorithm to extract the feature points of the rail track and select the appropriate curve model to construct the rail equation.

However, the method based on the feature extraction operator is relatively fixed to a specific scenario; it is difficult to apply one method in all railway environments. This is because the method based on the feature extraction operator is usually rule-based, while in the actual environment, the rail track is changeable, and the single rule is insufficient to describe the rail track in detail. Furthermore, with the rapid development of deep learning in recent years [19,20], the convolutional neural network has been applied in

many fields [21,22]. We find that the convolutional neural network has a strong feature extraction capability for the whole image. Segmentation networks such as SegNet [23] and ERFNet [24] show good performance in image segmentation. Chaurasia et al. [25] propose the ENet network structure by reducing the size of the feature map in advance and reducing the size of the decoder, which improved the running speed of the model. ROMERA et al. [24,26] designed the ERFNet network to improve the segmentation accuracy while keeping the model with low computational complexity. However, these methods are aimed at semantic segmentation of automotive autonomous driving, few studies focus on the semantic segmentation of rail track. Therefore, this study focuses on the semantic segmentation of rail track based on deep learning.

### 2.2. Obstacle Detection

In the research of obstacle detection technology, one may leverage the efficient existing methods based on deep learning models for object detection.

Object detectors based on CNN, such as Faster RCnn [22], SSD [27], and YOLO [28], can recognize and locate objects in images. Faster RCnn generates a large number of bounding boxes and then performs image recognition on each of them. YOLO directly predicts category scores and box offsets in multiple categories within one step, with a fixed set of default bounding boxes of different sizes at each position of the feature map with different shapes. SSD is a combination of Faster RCNN and YOLO; the regressor-based model is adopted to directly return the category and location of objects in the network, and the region-based concept is also used. In the process of detection, many candidate regions are used as ROI.

Based on the Caffe framework, Garnett [29] uses Google net as the backbone network and SSD and StixelNet as two network branches for pedestrian position detection and conventional obstacle detection, respectively. The networks use the labels generated by LiDAR for training and realize the detection and classification of three-dimensional information of objects through the network.

Aycard et al. [30] use the original data of LiDAR and stereo vision system as input to detect the surrounding obstacles in the visual sensing system and LiDAR sensing system, respectively, and obtain the list of moving objects. Premebida et al. [31] propose a pedestrian and vehicle detection system using cameras and LiDAR. The system uses a traditional Adaboost classifier and GMM classifier to classify ROI (region of interest) data and obtain detection results; however, the performance of the classifier has limited accuracy. Fan et al. [32] applied a 3D point cloud camera for object detection and distance measurement, but the algorithm is complex and demonstrates low real-time performance and the equipment is expensive. Garcia et al. [33] designed a detection architecture in which the vehicle is equipped with visual–LiDAR and LiDAR for demanding different distances. The method combines detection-level fusion with track-level fusion which improves the robustness and detection accuracy.

Current detection methods can only detect common obstacles within the detection range of sensors. However, the railway scene often involves unpredictable obstacles. At the same time, the current methods lack the understanding of the railway scene. Only the obstacles within the rail track limit will affect the operation of the current train, and the obstacles within the near rail track limit do not need the warning. Aiming at these shortcomings, in this paper, we propose an intelligent train detection system. The proposed system not only detects the obstacle, but also detects the rail track. The results of obstacle detection and rail track detection are further combined in order to judge whether the detected obstacle affects the train running.

### 2.3. Visual–LiDAR Fusion

The algorithms of visual–LiDAR fusion can be divided into pre-fusion and post-fusion:

(1)　Pre-fusion includes data fusion and feature fusion. The main method is to combine the features of the original image data and point cloud data, and then detect and

locate them. Rovid et al. [34] proposed a neural network method to fuse the image and LIDAR point cloud to form a dense point cloud and estimate the position of the target based on the point cloud. However, the pre-fusion method consumes a lot of resources and is insufficient to realize the on-board real-time application;

(2) Post-fusion is also called decision fusion. Firstly, the data obtained by the two sensors are detected, and the target detection results are fused and located. Sualeh et al. [35] first use the network to detect the image and point cloud, respectively, and find the two corners and the third corner of the diagonal of the target detection bounding box according to the three-dimensional point cloud so as to update the shape and position of the target.

Most of the existing rail track obstacle detection method relies on a single sensor; few of them are based on multi-sensor fusion. To detect the objects in front of the train quickly and obtain the shape and position of the target accurately. We use post-fusion technology to detect the shape and position of the obstacles.

## 3. Method

This section first introduces the workflow of the proposed method, and then each functional part will be introduced in detail.

### 3.1. Workflow

The main function of the intelligent train detection system is to detect the obstacles in the rail track and give early warning. The algorithm needs to provide information about the number, type, and distance of obstacles. To realize the sensing function, the intelligent train detection system in this paper includes the following equipment as the input: one short focal camera is used to detect near obstacles, and one long focal camera is used to detect long-range obstacles. Besides, a LiDAR is used to measure the distance of the obstacle.

The main components of the intelligent train detection system consist of rail region identification, obstacle detection, vision and LiDAR fusion, and spatial relationship judgment. The rail region identification module detects the train running area and provides rail region and rail track direction information; the obstacle detection module detects common obstacles in rail transit, such as trains and pedestrians, and provides location and classification information; the vision and LiDAR Fusion module detects the unknown obstacles in the rail region (moreover, it can output the distance information of obstacles which are detected by the obstacle detection module); the spatial relationship judgment module can judge whether the obstacle needs warning according to the rail track direction and the position of the obstacle. The workflow of the intelligent train detection system is shown in Figure 1.

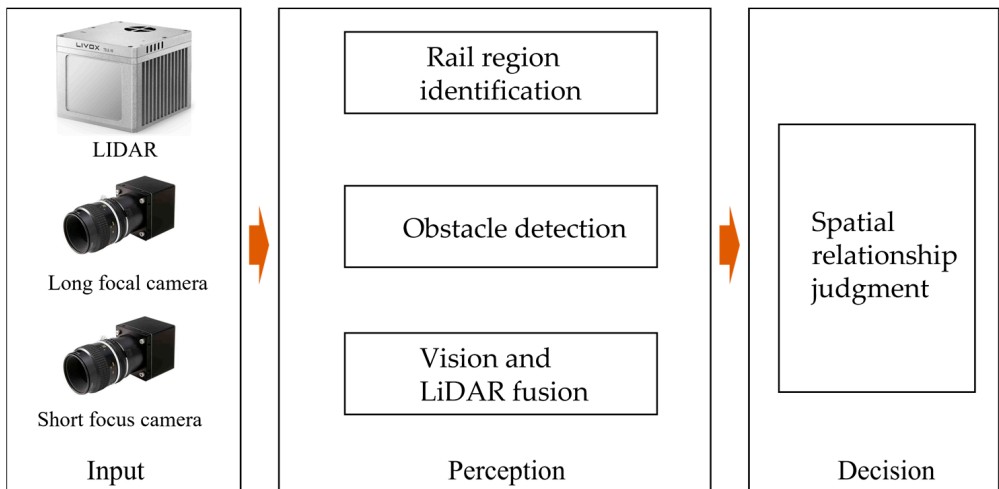

**Figure 1.** The workflow of the intelligent train detection system.

*3.2. Rail Region Identification*

One core function of the intelligent train detection system is rail track identification, which is to extract the rail area from the input image. Accurate and reliable rail region identification can not only provide accurate rail limits [36,37], but also accurate regions of interest [38,39] for obstacle detection. This makes obstacle detection more accurate and efficient. However, the traditional image processing algorithm is difficult to adapt to all scenes because of the complex environment of the rail region in the operation environment and the interlacing of various scenes such as curves and straights. Besides, in recent years, artificial intelligence, especially deep learning, has achieved rapid development and has been widely used in various fields. We find that the semantic segmentation algorithm has a strong ability to extract image features. Therefore we try to apply semantic segmentation to rail region recognition which mainly include a cascading downsampling layer (Cascading downsampling layer: Downsampling is performed by convolution and pooling), a convolution layer, a dilated convolution layer (Dilated convolution layer: Convolution with skip connections), a deconvolution layer (Deconvolution layer: The inverse operation of convolution which increases the resolution of the feature map), and a Softmax layer (Softmax layer: Used to achieve pixel-level classification). The image segment network structure is shown in Figure 2.

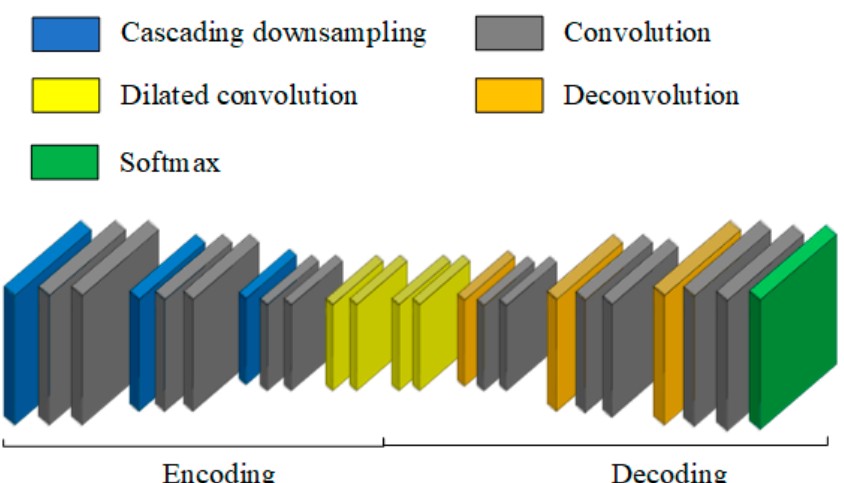

**Figure 2.** Architecture of image segment network.

The rail region recognition network mainly consists of the encoding section and decoding section. The encoding section is mainly used to analyze and extract the features of the rail region, which is composed of the convolution layer and the pooling layer. In addition, the encoding section also contains a dilated convolution layer. The convolution layer is mainly used to extract the rail region features of each layer. The pooling layer is mainly used to reduce the resolution of the image. On the one hand, the subsequent convolutional layer can have a larger receptive field. On the other hand, the computational amount of subsequent convolutional operations can be reduced to ensure the real-time performance of the algorithm. Dilated convolution extracts the features of different scales by applying different dilated convolution coefficients. These features are combined to achieve feature extraction of different scales without changing the size of the image feature map. The decoding section is followed by the encoding section. The decoding section mainly consists of a series of upsampling layers. The semantic information of the features extracted from the encoding section is restored to the original size to realize the pixel-level image understanding. In the last part of the network is the Softmax layer, which is mainly used to classify each pixel in the image to achieve pixel-wise rail track segmentation. The specific structure of the rail region detection network is shown in Table 1.

**Table 1.** Rail region detection network.

| Level | Type | Output | Feature Map Size |
|---|---|---|---|
| 1 | Cascading downsampling | 16 | $240 \times 180$ |
| 2–3 | Convolution | 64 | $240 \times 180$ |
| 4 | Cascading downsampling | 128 | $120 \times 90$ |
| 5–6 | Convolution | 64 | $120 \times 90$ |
| 7 | Cascading downsampling | 256 | $60 \times 45$ |
| 8–9 | Convolution | 256 | $60 \times 45$ |
| 10–11 | Dilated convolution (rate = 2) | 256 | $60 \times 45$ |
| 12–13 | Dilated convolution (rate = 4) | 256 | $60 \times 45$ |
| 14 | Deconvolution | 128 | $120 \times 90$ |
| 15–16 | Convolution | 128 | $120 \times 90$ |
| 17 | Deconvolution | 64 | $240 \times 180$ |
| 18–19 | Convolution | 64 | $240 \times 180$ |
| 20 | Deconvolution | 16 | $480 \times 360$ |
| 21–22 | Convolution | 16 | $480 \times 360$ |
| 23 | Softmax | 2 | $480 \times 360$ |

The segmentation results are shown in Figure 3. Green pixels are the track area.

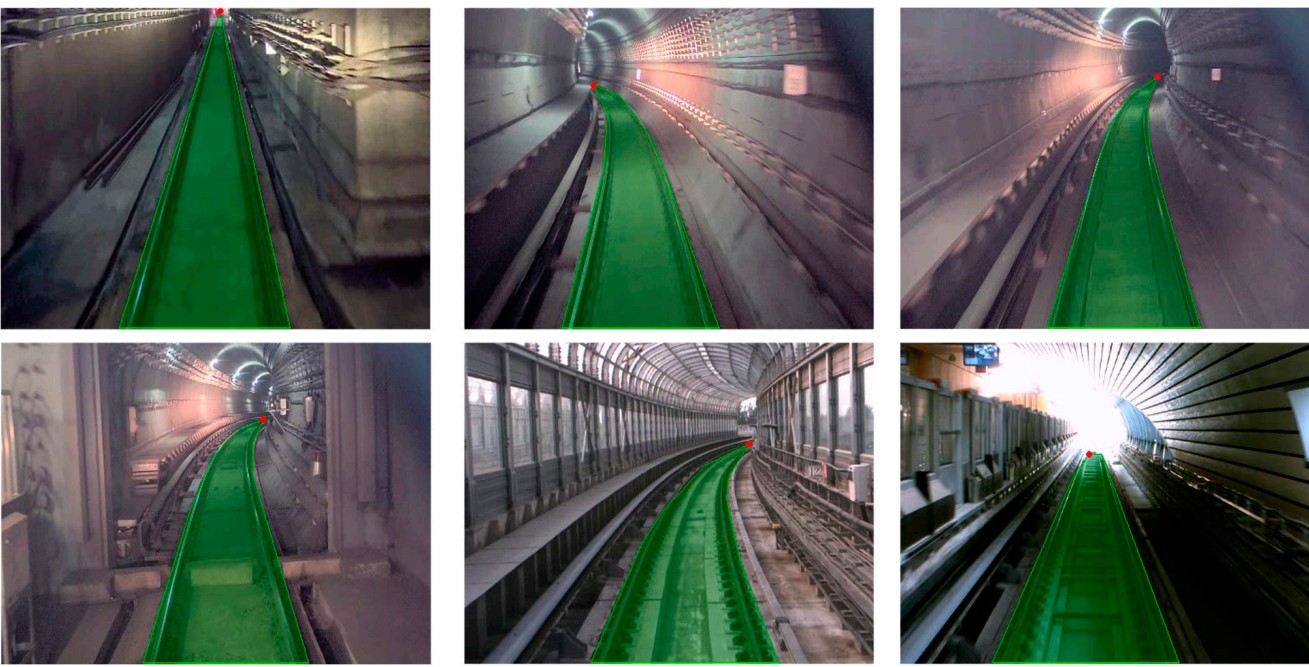

**Figure 3.** Rail area segmentation results.

### 3.3. Obstacle Detection

In obstacle detection part, the system uses a convolutional neural network to detect obstacles in the rail region. Depthwise separable convolution and residual network structure are used. We combine the output of the multi-layer feature layer to implement the detection, which includes bounding box regression and obstacle classification.

Considering the real-time requirements of the algorithm for practical train applications, the forward obstacle detection network was designed based on the fast deep learning objects detection network SSD [27]. The network includes three main parts: the first part is the feature extraction network, which extracts the features of the input rail environment image through multi-layer neural networks; the second part is the extra convolution layer, which aims to obtain the feature maps with a smaller scale; the third part is the network prediction layer, which is used to predict the target position of the forward obstacle. The obstacle detection network structure is shown in Figure 4. As the main part of the train

forward obstacle detection framework, the feature extraction network consumes most of the computing resources of the network. At the same time, the feature extraction of the rail environment image is also determined by this part. In the forward obstacle detection network, the residual connection is used in the design of the network [40]. The residual connection can eliminate the gradient vanishing. Meanwhile, depthwise separable convolution is applied in the forward obstacle detection network [20]. Depthwise separable convolution can reduce the computation of the network while maintaining the accuracy of the network. In addition, multiscale feature map fusion is applied in the design of a forward obstacle detection network. Multi-scale feature fusion combines feature maps with different resolutions so that the rich detail information in the lower layer of the network can be fused with the rich feature information in the upper layer. Therefore, the network has a stronger feature extraction ability.

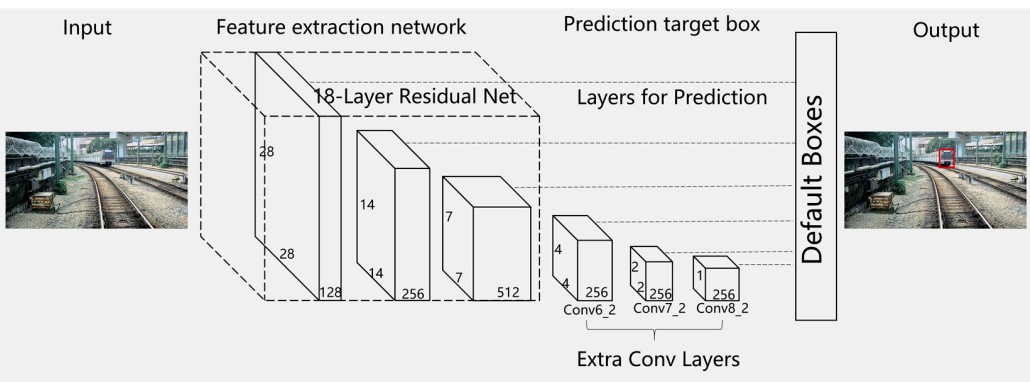

**Figure 4.** Image detection network structure.

The goal of setting extra layers in the train forward obstacle detection framework is to further compress the features extracted by the feature extraction layer to obtain more global features. Thus, it can be used for the subsequent detection of large-size obstacles.

After completing the design of the extra layer, the network prediction layer should be designed. The network prediction layer selects several layers from different feature graphs for decision-making. Each pixel in the feature map of the prediction layer can map its position to the original graph and corresponds to an obstacle detection bounding box of fixed size in the feature map. To detect obstacles of different sizes, six sizes of feature maps are selected for prediction. The category of the target bounding box at each position and the offset of the initial bounding box were obtained. The final size and position of the obstacle detection bounding box were obtained by combining the location information of the initial target bounding box set.

The a priori bounding box used to predict trains gradually increases from low-level feature maps to high-level feature maps. The formula used to predict the a priori bounding box of layer $k$ is shown in Equation (1):

$$S_k = S_{\min} + \frac{S_{\max} - S_{\min}}{m - 1}(k - 1), \ k \in [1, m] \tag{1}$$

where $S_k$ is the predicted bounding box the size of which is between the $S_{\min}$ and the $S_{\max}$, $S_{\min}$ is the minimum scale for prediction, $S_{\max}$ is the maximum scale for prediction, and m is the number of feature maps for prediction. In addition, we added three additional layers at the end of the train identification network to improve the accuracy of train detection. The specific structure of the additional layers is shown in Table 2.

The obstacle detection effect is shown from Figures 5–7. It can be seen that the proposed method has a good effect on a variety of scenarios such as daylight, night, tunnels, and elevated tracks.

**Table 2.** Structure of additional layers.

| Level | Convolution Kernel | Output |
|---|---|---|
| Conv6_1 | $1 \times 1 \times 128$ | $7 \times 7 \times 128$ |
| Conv6_2 | $3 \times 3 \times 256$ | $4 \times 4 \times 256$ |
| Conv7_1 | $1 \times 1 \times 128$ | $4 \times 4 \times 128$ |
| Conv7_2 | $3 \times 3 \times 256$ | $2 \times 2 \times 256$ |
| Conv8_1 | $1 \times 1 \times 128$ | $2 \times 2 \times 128$ |
| Conv8_2 | $2 \times 2 \times 256$ | $1 \times 1 \times 256$ |

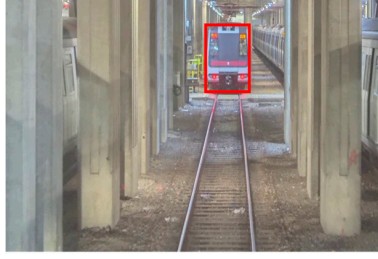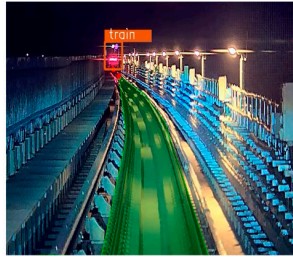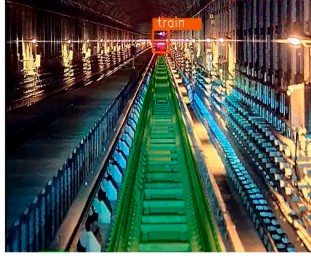

**Figure 5.** Detection results of straight road, curves and multi-turnout areas.

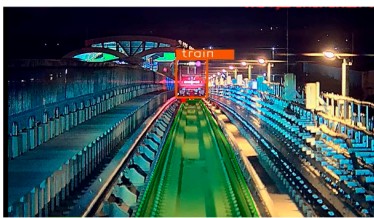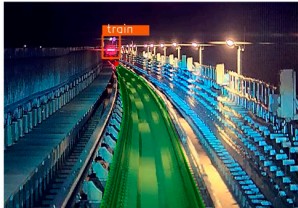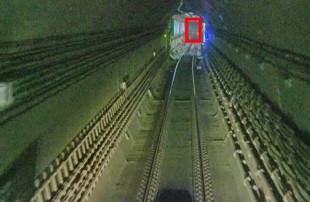

**Figure 6.** Detection results of night and tunnel conditions.

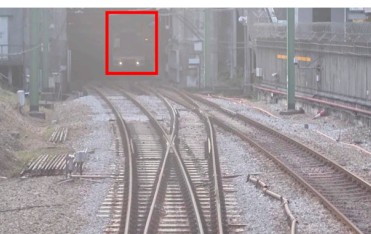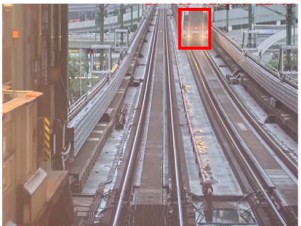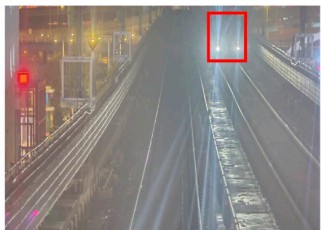

**Figure 7.** Detection results of bad weather.

*3.4. Visual–LiDAR Fusion*

The intelligent train detection system further combines the information of images and LiDAR sensors to achieve a better understanding of the environment. Firstly, the system unifies the visual and LiDAR data through combining the spatiotemporal information. Multi-level vision and LiDAR fusion algorithms are used to obtain the distance and class information of obstacles. Combined with prior knowledge, the results are checked again to obtain the final perceptual outputs.

Single visual information is difficult to achieve accurate obstacle distance measurement. In addition, visual information is susceptible to changes in light and other factors. In addition, a single LiDAR is difficult to determine the type of obstacles, and it cannot accurately identify the boundaries of the tracks. The intelligent train detection system integrates visual information with LiDAR information to realize accurate all-weather obstacle perception.

(1) Multi-sensor spatiotemporal fusion

The visual module of the intelligent train detection system is mainly used for rail area and train recognition. At the same time, the LiDAR module preprocesses the LiDAR

point cloud data and filters out the data with a large height of the point cloud. After the preprocessing of point cloud data, the spatial LiDAR points are mapped to the image through data association. Thus, sparse image depth information can be obtained.

(2) Abnormal obstacle screening

In the actual train operation environment, there are some objects which rarely appear, but the occurrence of them will have an impact on the train operation. Therefore, we need to detect these abnormal obstacles. "Abnormal obstacle" refers to an object with a small probability. Due to insufficient samples, these objects are difficult to be learned by the deep learning method. After obtaining the depth information of the image, the rail region LiDAR points are extracted by mapping of the rail region image. The rail surface fitting of the LiDAR point cloud is carried out by the RANSAC [41] algorithm to obtain the space equation of the rail surface. Based on the space equation of the rail surface, the point clouds above the rail plane are clustered to identify the obstacles within the rail boundaries.

(3) Distance measurement

According to the mapping relationship between the image and LiDAR, the LiDAR point corresponding to the obstacle detected by the image is determined. The average distance of the LiDAR point set to the obstacle is calculated as the distance of the obstacle. The fusion effect of the image and LiDAR is shown in Figure 8.

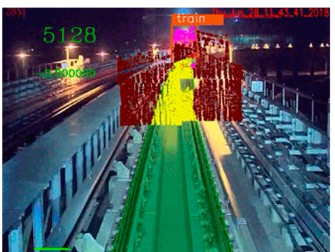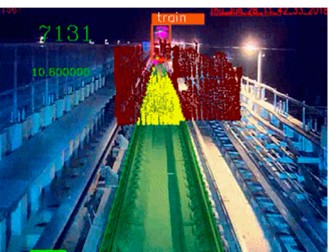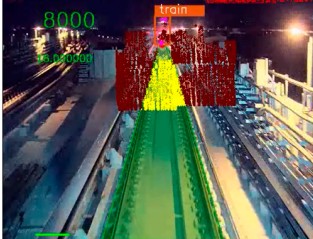

**Figure 8.** Illustration of image and LiDAR fusion.

*3.5. Spatial Relationship Judgment*

During the actual train operation, only the obstacles on the train track will affect the train operation. On the contrary, the obstacles outside the rail track do not affect the train operation. For the intelligent train detection system, only the obstacles that have an impact on the normal operation of the train ahead need to be warned. The current algorithm only detects the obstacle and rail region but does not judge the relationship between the obstacle and the current train. Therefore, it is necessary to judge the relationship between the current rail and obstacles. The workflow of spatial relation judgment is shown in Figure 9.

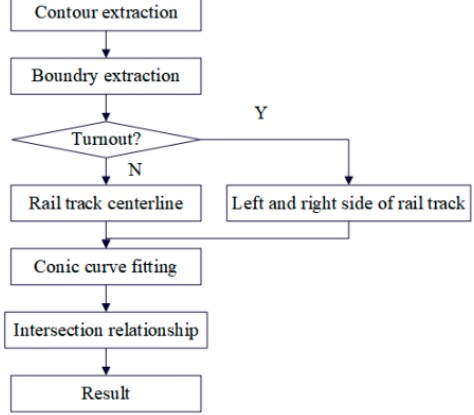

**Figure 9.** Workflow of spatial relation judgment.

Step1: Use the chain code tracking algorithm to extract the contours of the rail region [42] to obtain the outer contour of the rail region;

Step2: Select the set of key points on the left and right sides of the contour of the rail region as the left and right boundaries;

Step3: Determine whether there is a turnout core point according to the ordinate change of the outer contour set. If the ordinate of the outer contour set decreases and then increases, this point is the turnout core point. If there are turnout cores, proceed to 3.1; otherwise, proceed to 3.2;

Step3.1: Use the quadratic function to fit the left and right contour sets, respectively, and get two rail curves;

Step3.2. Use the quadratic function to fit the midpoints of the left and right contours to obtain the rail center curve;

Step4: If the rail curve intersects the lower box line of the obstacle bounding box, the obstacle is considered to be on the current rail track; otherwise, the obstacle is on the adjacent rail track.

## 4. Experiment Results

The experiment was conducted on Hong Kong Tsuen Wan line. There are 16 stations along the whole of the Tsuen Wan Line, with a total length of about 16 km. Among them, 4 stations of Tsuen Wan Station, Kui Hing Station, Kui Fong station, and Li Jing Station are above-ground stations, and the other 12 stations are underground. The operation line includes the ground section and tunnel section. The smallest radius of the line is 310 m and the maximum grade of the line is 3%. It is a typical rail transit line, which can cover most scenes encountered by train operation.

To evaluate the effectiveness of the proposed method, the test lasted nine months and involved the installation of Train Intelligent Detection System (TIDS) equipment on five trains of the Tsuen Wan Line as well as the corresponding on-time running tests, with a cumulative mileage of more than 200,000 km and a total of more than 50,000 km of data collected after installation. The video with resolution of 1280 × 720 are collected. All experiments were trained based on Titan Xp GPU, Pytorch 1.1.0, CUDA 9.0, and CUDNN 7.1.

For a straight track, TIDS can achieve more than 240 m detect range. The image detection effect is shown in Figure 10. For a grade 3% maximum ramp, TIDS can achieve no less than 100 m detection. The detection effect is shown in Figure 11.

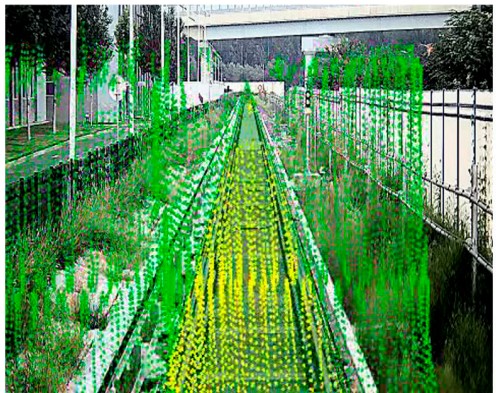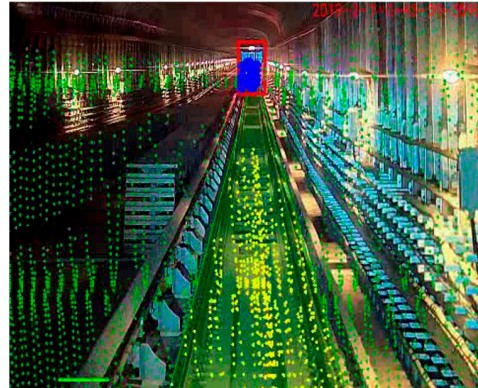

**Figure 10.** Straight track detection.

In the case of the sharpest curve of the Tsuen Wan Line (R = 312 m), the system can achieve a detection distance of no less than 70 m. The image detection effect is shown in Figure 12.

We sampled more than 100 h of data for statistical analysis of false alarm rate (FAR), miss detection rate (MDR), perception, and recall of train obstacle detection system results. The results of this analysis are shown in Table 3.

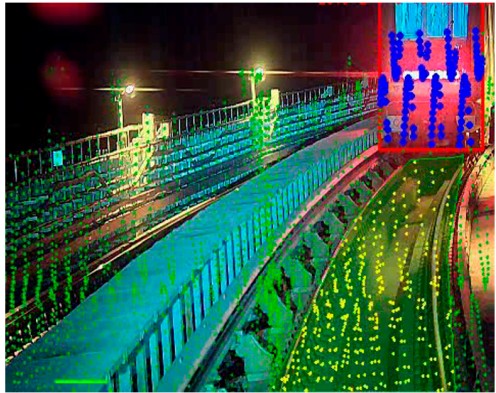 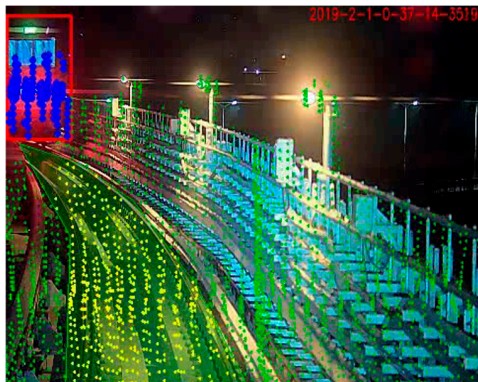

**Figure 11.** Ramp detection.

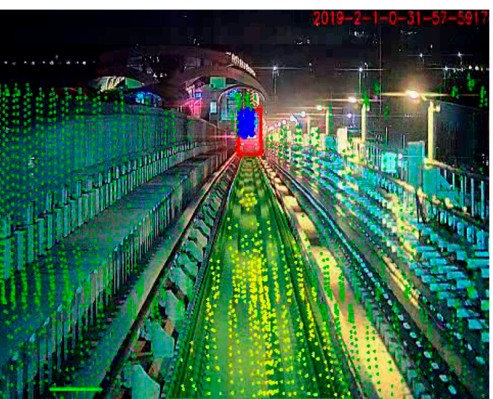 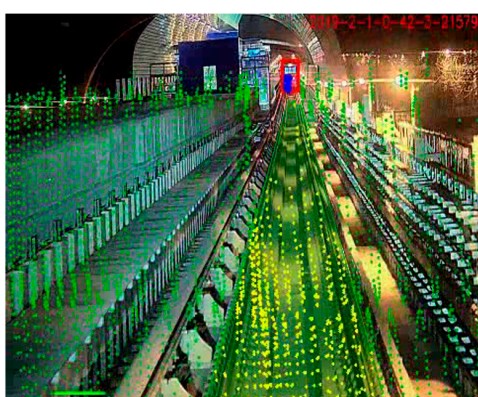

**Figure 12.** Curve (R = 312 m), maximum visible distance 73 m.

**Table 3.** Analysis of experimental results of intelligent train detection system.

| The System Time | Results |
| --- | --- |
| Effective running seconds (s) | 363,268 |
| False alarm time(s) | 30 |
| Miss detection time(s) | 0 |

Table 4 shows the analysis results and the detection indicators of this system. The false alarm rate is 0.008% and the miss detection rate is 0. It shows that our method has higher precision. Although there are still some false alarms in our methods at this stage, as a perception system, false alarms do not cause security problems. On the contrary, it is miss detection that can cause the safety problems, and the miss detection rate is 0 in this study. This shows that our method can meet the requirements of practical application.

**Table 4.** Indicator of intelligent train detection system.

| Indicators | Results |
| --- | --- |
| False alarm rate | 0.008% |
| Miss detection rate | 0 |

At the same time, to prove the effectiveness of the fusion method proposed in this paper, we used vision and LiDAR for train detection, respectively. For the proposed fusion method, we can achieve accurate forward obstacle detection and distance measurement. The forward obstacle detection abilities of different methods are shown in Table 5.

**Table 5.** Forward obstacle detection ability.

| Obstacle Distance (m) | Metrical Error | | |
|---|---|---|---|
| | **Vision Method** | **LiDAR Method** | **Fusion Method** |
| 60 | −4% | −1.83% | −0.5% |
| 80 | 2.75% | 1% | 0.75% |
| 100 | 3.7% | × | 1.1% |
| 120 | 3.5% | × | −0.25% |
| 140 | 5.5% | × | 1.57% |
| 160 | 5.1% | × | 2.69% |
| 180 | 6.28% | × | −1.44% |
| 200 | 6.8% | × | 1.45% |
| 220 | 9.82% | × | −1.4% |
| 240 | 7.59% | × | 1.46% |

The "×" in Table 5 indicates that the obstacle cannot be detected. The results show that our fusion method can achieve better obstacle detection than the single sensor method. This is because we know the pixel of the forward obstacle from the camera and its corresponding point from LiDAR; thus, we can achieve accurate forward obstacle detection and distance measurement.

## 5. Conclusions

In this paper, an intelligent train detection system is proposed and applied to the urban rail scenario. The application of different focal cameras improves the perceived range. The fusion of LiDAR supplements the distance information which is missing from the image and realizes the detection of unknown small obstacles. In addition, the proposed system makes full use of existing deep learning-based object detection and semantic segmentation algorithms to extract regions of interest based on subway scene features. Finally, the spatial judgment algorithm meets the perception requirements of the rail scenario. Experiments show that the system can detect the obstacles on the rail track reliably and robustly.

As the active obstacle detection technology is increasingly widely accepted by the market, the key technologies and implementation methods of the TIDS system must be paid more attention. In the future, this technology will develop in the direction of high RAMS. When the FAO line is operating without a driver on duty, the highly reliable SIL4 TIDS will serve as the "eyes" of the train to further detect foreign objects intruding into the boundary in real time and report to the signal system [43,44]. The existing methods mainly focus on on-board obstacle detection; however, the braking distance of the train is too long, and the autonomous perceived distance of the train cannot meet the braking requirements in curves and other scenarios. Therefore, in the future, it is necessary to further develop vehicle–road collaborative perception and improve obstacle detection distance. In addition, a convolutional neural network is used in this study; neural networks have uncertainty, especially in a dynamic environment [45,46]. In future studies, we will estimate the uncertainty to improve the interpretability of the model.

**Author Contributions:** Conceptualization, Q.Z. and F.Y.; methodology, Q.Z.; software, R.W.; validation, W.S.; formal analysis, R.W.; investigation, R.W.; resources, G.L.; data curation, F.Y.; writing—original draft preparation, Q.Z.; writing—review and editing, Q.Z.; visualization, G.L.; supervision, F.Y.; project administration, F.Y.; funding acquisition, F.Y. All authors have read and agreed to the published version of the manuscript.

**Funding:** This research was funded by Guangxi Key Research and Development Program, grant number AB22035008 and the Basic Research Funds for Central Universities, grant number 2022JBZY024.

**Institutional Review Board Statement:** Not applicable.

**Informed Consent Statement:** Not applicable.

**Data Availability Statement:** Data is not available for commercial reasons at this stage.



**Acknowledgments:** The authors would like to thank the insightful and constructive comments from anonymous reviewers.

**Conflicts of Interest:** The authors declare no conflict of interest.

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
