# Peer review of "Automatic Obstacle Detection Method for the Train Based on Deep Learning"

_sustainability, doi:10.3390/su15021184_

Round 1

Reviewer 1 Report

Ln. 56. The following has been written, 'This solution is economical, simple to deploy, and can be employed in a dark environment'. The reason why it is economical and simple to deploy should be elaborated. The statement requires a few citations to show your reader you've done proper research by listing sources you used to get your information. 

Ln. 69 to 72. The whole paragraph is wrong. Delete them all. It has long been adopted for many railway-infrastructure. Just an example for level crossing; https://lbfoster.eu/en/case-studies/control-and-display/network-rail-level-crossing-obstacle-detection-systems/

Ln80 to 85, Do not use the first-person point of view, i.e., we. 

Ln.87. Change it to Literature Review

Ln.100. The following has been written, 'However, the method based on the feature extraction operator is relatively fixed to a specific scenario, so it is difficult to apply one method in all railway environments.' Elaborate on why this is so. 

Ln.114. Explain somewhere SSD in the paragraph as done for Faster RCnn and YOLO. 

Ln. 134. Revise the sentence. 

Ln. 141. Give a hint here about an intelligent train detection system that is proposed. 

Ln. 179. The paragraph attempts to explain the steps shown in Figure  1. One helpful approach to writing research articles is to list each of your key findings in a logical order. Then, create figures or tables illustrating the data that you intend to report. This, in turn, provides the skeletal content and structure of your Methods section, which should match the order in which the figures are provided. However, the steps in the figure are not logically explained.

Please use footnotes for the terms to explain; Cascading downsampling, Convolution, Dilated convolution, Deconvolution, and Softmax. 

Ln.199. Explain why the chosen order in levels 1 to 23 in Table 1 is correct. 

Ln. 217. Use high-res pictures. 

Ln. 245 Elaborate Feature extraction network and Prediction target box in detail. 

Ln.307 Please magnify them all (Figure 8). 

Please write a limitation underlying the weaknesses of the study, based on factors that are often outside of your control as the researcher.

Author Response

The authors would like to thank the reviewer for their insightful and constructive comments. The original manuscript has been thoroughly revised and improved by addressing these comments and suggestions. The item-to-item responses to each issue indicated in the review report are provided in the attachment.

Reviewer 2 Report

The author has done a study on the autonomous operation of urban rail trains mainly depending on the signalling control system. However, it lacks extra equipment to percept the environment for active protection. To further enhance the efficiency and safety of the widely deployed fully automatic operation (FAO) systems of urban rail, this study proposes an intelligent urban rail obstacle detection system based on deep learning. It collects perceptual information from the industrial cameras and Light Detection and Ranging (LiDAR), and mainly implements functionality including rail region detection, obstacle detection, and visual-LiDAR fusion. Specifically, the first two parts adopt deep convolutional neural network (CNN) algorithms for semantic segmentation and object detection to pixel-wisely identify the rail track area ahead and detect the potential obstacles on the rail track, respectively. The visual-LiDAR fusion part integrates the visual data with the LiDAR data to achieve environmental perception for all weather conditions. It can also determine the geometric relationship between the rail track and obstacles to decide whether to trigger a warning alarm. Experimental results show that the system proposed in this study has strong performance and robustness. The system perception rate (precision) is 99.994% and the recall rate reaches 100%. The system improves the safety of urban rail train operation effectively and is applied in the metro Hong Kong Tsuen Wan line.

Find below some aspects that authors must address:

1.       The authors must clearly state the novelty and main contributions of this work when compared with a large amount of literature available over the last decades.

2.       Need to improve figure resolution.

3.       The Rail track region detection modelling describes in Section 2 is relatively simple. Better justify the use of this particular model.

4.       Describe the “experiment conducted on the Hong Kong Tsuen Wan liner, and describe the reason for selecting the particular line.

5.       Clarify how the experiment is conducted. how about the numerical and computational aspects?

6.       Comment on the computational/experimental accuracy and efficiency of the proposed approach.

7.       The results are sound; anyway, it would be great if the author could clarify the benefits of the presented approach when compared with some alternative model/software.

8.       The “Conclusions” section must be expanded.

9.   Kindly improve the English of the article.

Final comment: Major revision.

Author Response

(The authors gave the same response as above.)

Reviewer 3 Report

Focus of the paper is important for the real-life problems, but its academic side should be improved significantly. Please consider the comments below to improve the paper further:

1)      Please provide more accurate and informative title for the paper. The title should reflect the key contributions of the paper. Please re-consider “urban rail” in the title.

2)      English of the paper should be improved as well.

3)      Abstract of the paper should be improved. The first sentence can state the importance of the content, then the gaps in the corresponding literature. Key contributions of the paper should be expressed clearly and then the major findings of the paper should be provided.

4)      Introduction has provided some background researches and highlighted their advantages and disadvantages. However, critical review of the recent and related works are not quite strong. The corresponding gaps should be emphasized strongly and based on these gaps, the claimed contributions of the paper should be justified. Contributions must be academically sound and clear.

5)      Performing a comparison-based analyses with a recent and related approach under the equal conditions could help to improve and justify the contribution of the paper.

6)      Please improve the equations by adding brief insights about them. In the current form they are raw.

7)      Please specify the kind of uncertainties. They can be internal or external, parametric or non-parametric, constant, characteristic or random. Determining their structures and amounts are challenging in the real time applications.

8)      What are the possible problems that the proposed algorithm can face in real time applications? What are the physical, mechanical, electrical and environmental constraints which are unavoidable in real time environments? 

9)      Forming a constrained optimization problem in an unstructured environment under the dynamic environment conditions should be addressed properly. In addition, recent and related literature should be examined thoroughly. The authors are suggested to see these recent and related papers:  Completely Model Free Adaptive Control Design in Presence of Parametric, Non-parametric Uncertainties and Random Control Signal Delay. CPG Based RL Algorithm Learns to Control of Humanoid Robot Leg. The first paper considers various uncertainties and delays to control an autonomous system. The second one develops a central pattern generator based intellignet control approach to handle the desired control of  ahigher robot leg.

10)  Is it “intelligent train detection” or “vision based obstacle detection for the trains”?

11)  Please explain how the “abnormal obstacle” is defined for the CNN?

12)  Please state what “spatial relationship jedgement” is. Why spatial and how is it formulated?

13)  Please justif Equation 1.

14)  Figures and sections should be improved.

Good luck with the improvements…

Author Response

(The authors gave the same response as above.)

Round 2

Reviewer 3 Report

The paper has been revised properly and can be accepted with the current version.